# CHARACTERIZING ATTACKS ON DEEP REINFORCEMENT LEARNING

## ABSTRACT

Deep reinforcement learning (DRL) has achieved great success in various applications, such as playing computer games and controlling robotic manipulations. However, recent studies show that machine learning models are vulnerable to adversarial examples, which are carefully crafted instances that aim to mislead these models to make arbitrarily incorrect prediction. DRL models have been attacked by adding perturbations to observations. While such observation based attack is only one aspect of potential attacks on DRL, other forms of attacks require further analysis which are more practical such as manipulating environment dynamics. Therefore, we propose to understand the vulnerabilities of DRL from various perspectives and provide a thorough taxonomy of adversarial attacks against DRL. We also conduct the first set of experiments on the unexplored parts within the taxonomy. In addition to current observation based attacks against DRL, we propose attacks based on action space and environment dynamics. Among these experiments, we introduce online sequential attack: a novel method specialized for efficiently attacking sequences of temporally consecutive frames, the first targeted attack that perturbs environment dynamics to cause an agent fail in a specific way, and a novel finite difference based black-box attack against DRL. We show empirically that our online sequential attacks can generate effective perturbations in a black-box setting in real time with a small number of queries, independent of episode length. We conduct extensive experiments to compare the effectiveness of different attacks with several baselines in various environments, including game playing, robotics control, and autonomous driving.

## 1 INTRODUCTION

In recent years, deep neural networks (DNNs) have become pervasive and led a trend of fast adoption in various commercial systems performing image recognition (Krizhevsky et al., 2012), speech recognition (Hannun et al., 2014), and natural language processing (Sutskever et al., 2014). DNNs have also encouraged increased success in the field of deep reinforcement learning (DRL), where the goal is to train an agent to interact with the environments for maximizing an expected return. DRL systems have been evaluated on games (Ghory, 2004; Mnih et al., 2013; 2016), autonomous navigation (Dai et al., 2005), and robotics control (Levine et al., 2016), etc. To take advantage of this, industries are integrating DRL into production systems (RE WORK, 2017). However, it is well-known that DNNs are vulnerable to adversarial perturbations (Goodfellow et al., 2015; Li & Vorobeychik, 2014; 2015). DRL systems that use DNNs to perform perception and policy making also have similar vulnerabilities. For example, one of the main weaknesses of RL models in adversarial environments is their heavy dependence on the input observations. Since RL models are trained to solve sequential decision-making problems, an attacker can perturb multiple observations. In fact, the distribution of training and testing data could be different due to random noise and adversarial manipulation (Laskov & Lippmann, 2010). Therefore, the learned policy can be vulnerable in adversarial environments.

In this paper, we first present an extensive study of adversarial attacks on DRL systems. Second, we propose and evaluate 10 adversarial attacks in order to explore points in the taxonomy that have not previously been examined in the literature. We organize adversarial attacks on DRL into a taxonomy based on details of the victim model and other properties of the attacker. First, we categorize these attacks based on **what component of the system** the attacker is capable of perturbing. The organization of this categorization resembles the components of a Markov decision process (MDP): we recognize attacks that perturb an agent's *observations*, *actions*, or the system's *environment*

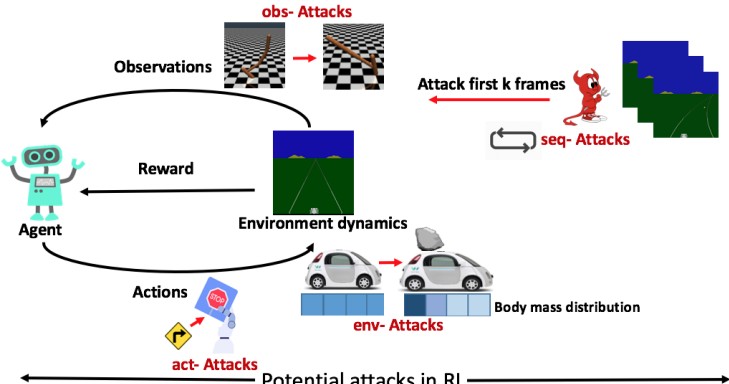

Figure 1: Taxonomy of adversarial attacks on deep reinforcement learning (DRL). RL environments are usually modeled as a Markov Decision Process (MDP) that consists of *observation* space, *action* space, and *environment (transition) dynamics*. Potential adversarial attacks could be applied to any of these components.

*dynamics*. We summarize these categories in Figure 1. Second, the **attacker's knowledge**. We categorize these attacks based on what knowledge the attacker needs to perform the attack. Broadly, this breaks attacks down into the already recognized *white-box* attacks, where the attacker has full knowledge of the target DRL system, and *black-box* attacks, where the attacker has less or no knowledge. We will discuss this taxonomy further in Section 3.

On the other hand, existing attacks that perturb the observation operate independently on each frame, which are too computational intensive to run in real-time. We propose two novel strategies for *quickly* creating adversarial perturbations to use in real-time attacks. The first strategy, *N-attack*, trains a neural network to generate a perturbation, reducing the computation to a single forward pass over this network. The second strategy exploits the property that, in RL environments, the states are not independent, and later states depends on previous state and action. Therefore, we propose *online sequential attacks*, which, in contrast to attacks that operate independently on each frame, generate a perturbation using information from a few frames and then apply the generated perturbation to later frames. We include our experiments with these strategies as part of our exploration of attacks that perturb the observation. We describe our attacks in detail in Section 4, and we present our evaluation in Section 5.

To summarize, our contributions are: (1) We systematically organize adversarial attacks on DRL systems into a taxonomy and devise and evaluate 10 new attacks on several DRL environments; (2) We propose two practical strategies, N-attack and online sequential attack, for performing real-time attacks on DRL systems; (3) We propose the attack that adversarially perturbs a DRL system's environment dynamics, and the attack that perturbs a DRL system's actions; (4) We apply finite difference method to estimate gradient and perform black-box attack, and we propose an adaptive sampling based method to improve the gradient estimation efficiency; (5) We propose to select the optimal frames within consecutive frames to attack and achieve optimal attack efficiency. We also provide corresponding mathematical analysis.

## 2 RELATED WORK

**Adversarial attacks on machine learning models**. Our attacks draw some of their techniques from previously proposed attacks. (Goodfellow et al., 2015) describes the fast gradient sign method (FGSM) of generating adversarial perturbations in a white-box setting. (Carlini & Wagner, 2017) describes additional methods based on optimization, which result in smaller perturbations. (Moosavi-Dezfooli et al., 2017) demonstrates a way to generate a "universal" perturbation that is effective on a set of multiple inputs. (Evtimov et al., 2018) shows that adversarial examples can be robust to natural lighting conditions and viewing angles. In our black-box attacks, we apply techniques that have been proposed for adapting white-box methods to black-box scenarios (Papernot et al., 2017; Chen et al., 2017). Recently, (Huang et al., 2017) demonstrates an attack that uses FGSM to perturb observation frames in a DRL setting. They experiment with a white-box attack and a black-box attack based on transferability. In this work, we propose novel black-box attacks, including the attacks

that do not rely on transferability. In addition, we propose several ways to reduce the computational complexity of attacks. (Lin et al., 2017) designs an algorithm to achieve targeted attack for DRL models. However, their work only considers targeted attack and requires training a generative model to predict future states, which is already a computational intensive task. (Behzadan & Munir, 2017) proposes a black-box attack method that trains another DQN network to minimize the expected return while still using FGSM as the attack method.

**Robust RL via adversarial training**. Safety in various robotics and autonomous driving applications has drawn lots of attention for training robust models. Knowing how RL models can be attacked is beneficial for training robust RL agent. (Pinto et al., 2017) proposes to train a RL agent to provide adversarial attack during training so that the agent can be robust against dynamics variations. However, since they manually selected the perturbations on environment dynamics, the attack provided in their work may not be able to generalize to broader RL systems. Additionally, their method relies on an accurate modeling of the environment dynamics, which may not be available for real world tasks such as robotics and autonomous driving systems.

## 3    TAXONOMY OF ATTACKS IN DRL

Existing work on attacking DRL systems with adversarial perturbations focuses on perturbing an agent's observations. This is the most appealing place to start, with seminal results already suggesting that recognition systems are vulnerable to adversarial examples (Goodfellow et al., 2015; Moosavi-Dezfooli et al., 2017). It naturally follows that we should ask whether perturbations introduced in other places in a RL system can cause the agent to misbehave, and in what scenarios, taking into account (i) can the attacker perform the attack with limited knowledge about the agent, (ii) can the attacker perform the attack in real time, and (iii) can the attacker introduce the perturbation physically. To systematically explore this question, we propose a taxonomy of possible adversarial attacks.

**Attack components**. In the first layer of our proposed taxonomy, we divide attacks based on what components in an MDP the attacker chooses to perturb: the agent's *observations*, *actions*, and *environment dynamics*. We will discuss some of the scenarios where attacks on these components can be practical. For attacks applied on the observation space, the pixel values of images can be changed by installing some virus into the software that is used to process captured photos from the sensor or in the simulator that is rendering the environment. In case images are transmitted between robots and computers, some communications can be altered by an attacker wirelessly (Lonzetta et al., 2018). Some physical observation based attacks have been analyzed in autonomous driving (Evtimov et al., 2018). For attacks applied on the action space, the action outputs can be modified by installing some hardware virus in the actuator executing the action. This can be realistic in some robotic control tasks where the control center sends some control signals to the actuator, a vulnerability in the implementation, for example, vulnerability in the bluetooth signal transmission, may allow an attacker to modify those signals (Lonzetta et al., 2018). For attacks applied on the environment dynamics, in the autonomous driving case we can change the material surface characteristic of the road such that the policy trained in one environment will fail in the perturbed environment; in the robotic control case, the robot's mass distribution can be changed such that the robot may lose balance when executing its original policy because the robot hasn't been trained in this case.

**Attacker's knowledge**. In the second layer of our proposed taxonomy, we categorize attacks based on what information the attacker needs to perform the attack. This divides attacks into *white-box* attacks and *black-box* attacks. We make a further categorization based on the attacker's knowledge about the policy network's architecture, weight parameters and whether the attacker can query the network. In **white-box** attacks, the agent has access to the architecture and weight parameters of the policy network and of course can query the network. In **black-box** attacks, the attackers don't have access to weight parameters of the policy network and may or may not have access to the policy network's architecture. The attacker may or may not have access to query the policy network.

**Further categorization**. We consider these additional properties of attacks. **Real-time**: while some attacks require more computation than can be performed in real-time, some are fast enough to run. Still other attacks perform some precomputation and then are able to generate perturbations quickly for each step. We identify this pragmatic property as part of our taxonomy. **Physical**: for RL tasks that take place in the real world, this property concerns the feasibility of physically applying the perturbation on the environment. **Temporal dependency**: we distinguish between attacks that

| Attack | MDP Component | Attacker Knowledge | | | | Real-time | Physical | Temporal Dependency |
|---|---|---|---|---|---|---|---|---|
| | | White/Black-Box | Arch. | Param. | Query | | | |
| obs-fgsm-wb | Observation | White-box | Yes | Yes | Yes | Yes | No | Independent |
| **obs-cw-wb** | Observation | White-box | Yes | Yes | Yes | Too slow | No | Independent |
| **obs-nn-wb** | Observation | White-box | Yes | Yes | Yes | Yes | No | Independent |
| obs-fgsm-bb | Observation | Black-box | No | No | No | Yes | No | Independent |
| **obs-imi-bb** | Observation | Black-box | No | No | Yes | Yes | No | Independent |
| **obs-fd-bb** | Observation | Black-box | No | No | Yes | Too slow | No | Independent |
| **obs-sfd-bb** | Observation | Black-box | No | No | Yes | Too slow | No | Independent |
| **obs-seq-fgsm-wb** | Observation | White-box | Yes | Yes | Yes | Yes | No | Sequential |
| **obs-seq-fd-bb** | Observation | Black-box | No | No | Yes | Yes | No | Sequential |
| **obs-seq-sfd-bb** | Observation | Black-box | No | No | Yes | Yes | No | Sequential |
| **act-nn-wb** | Action | White-box | Yes | Yes | Yes | Yes | No | Independent |
| **env-search-bb** | Dynamics | Black-box | No | No | Yes | N/A | Yes | N/A |

Table 1: Summary of the adversarial attacks on DRL systems, categorized based on our proposed taxonomy. The name reflects the category of the attack method. For example, **obs-nn-wb** means attack on observation using neural network based white-box attack. The attack methods we proposed are highlighted using bold text. "Arch.," "Param.," and "Query" indicate whether the attack requires knowledge of the policy network's architecture, parameters and whether it needs to query the policy network.

generate a perturbation in each frame *independently* from other frames and *online sequential attacks* that use information about from previous frames to generate perturbations on later frames.

# 4 ADVERSARIAL ATTACKS ON REINFORCEMENT LEARNING POLICIES

In order to study the unexplored parts of our proposed taxonomy from Section 3, in this section we develop several concrete attacks.

## 4.1 ATTACKS ON STATE OBSERVATIONS

We now describe attacks that perturb an agent's state observations. In this category of attacks, the attacker changes the input state observation $s$ to $\tilde{s} = s + h(s; w)$, where the attacker generates perturbation $h(s; w)$ from the original observation $s$ and some learned parameters $w$. In order to ensure that perturbations are small, we require that $||h(s; w)||_\infty \leq \epsilon$, which we can enforce by choosing $h$ to be of the form $\epsilon \tanh(\cdot)$. We present both white-box attacks and black-box attacks.

### 4.1.1 WHITE-BOX ATTACKS

In this setting, we assume that the attacker can access the agent's policy network $\pi(a|s)$ where $a$ refers to the action and $s$ refers to the state. (Huang et al., 2017) has previously introduced one attack in this category that applies the FGSM method to generate white-box perturbation purely on observations. We reproduce this experiment with our **obs-fgsm-wb** attack. This attack's application scenario is when we know the policy network's architecture and parameters. We also include a variant of Huang et al.'s attack that replaces FGSM with an optimization based method (Carlini & Wagner, 2017) in **obs-cw-wb**. In addition, we propose an attack strategy *N-attack* where the perturbation $h(s, w)$ is computed from a deep neural network. in a white-box setting. We call this attack **obs-nn-wb**. This attack works when where we know the policy network's architecture and parameters. We train the parameters $w$ of the attacker network based on the given policy $\pi$ to minimize victim policy's expected return when the perturbations are applied: $w = \arg\max_w \mathbb{E}_{\pi(a|\tilde{s})}[\sum_t \gamma^t \tilde{r}_t] = \arg\max_w \mathbb{E}_{\pi(a|s+h(s,w))}[-\sum_t \gamma^t r_t]$. With a fixed victim policy $\pi$, this attack is similar to training a policy. We include training details of this attack in Appendix B.

### 4.1.2 BLACK-BOX ATTACKS

In general, the trained RL models are kept private to avoid easy attacks. Given such "black-box" models, the adversary needs to take more sophisticated strategies to perform the attacks. In the black-box attack, there are different scenarios based on the knowledge of attacker. First, the attacker is not allowed to obtain any information about the model architecture, parameters, or even query information. In this case, the attacker can perform a "transferability" based attack by attacking a surrogate model and then transfer the perturbation to the victim model. Huang et al. introduced a black-box variant of the FGSM attack using transferability, which we denote as **obs-fgsm-bb**. This attack requires access to the original training environment. In this section, we introduce several other novel black-box attack methods and propose to improve the efficiency of these attacks.

**Imitation learning based black-box attack**. This attack **obs-imi-bb** is inspired by Rusu et al.'s work on policy distillation (Rusu et al., 2015). The attacker trains a surrogate policy $\hat{\pi}(a|s, \theta)$

to imitate the victim policy $\pi$. Then the attacker uses a white-box method on the surrogate policy to generate a perturbation and applies that perturbation on the victim policy. We include details description of this attack in our Appendix B.

**Finite difference (FD) based black-box attack**. Previous black-box attacks obs-fgsm-bb and obs-imi-bb all require retraining a surrogate policy. Previous work in (Bhagoji et al., 2017) applies the finite difference (FD) method in attacking classification models. We extend the FD method to DRL systems in obs-fd-bb which doesn't require retraining a new policy. This attack works in the setting where we don't have the policy network's architecture or parameters, but can query the network. We put the math details of FD method in Appendix B. For $n$ dimensional input, the finite difference method would require $2n$ queries to obtain the estimation, which is computationally intensive for high dimensional inputs such as images. We propose a sampling technique to mitigate this computational cost.

**Adaptive sampling based FD (SFD)**. Considering the fact that many deep learning models extract features from inputs patch-wise and have sparse activation map (Bau et al., 2017), we propose to iteratively estimate the gradients of high dimensional inputs and apply adaptive sampling to sample candidate dimensions for querying. Given a function $f(X;w) : X \to Y$, where $X \in \mathbb{R}^d$ and $Y \in \mathbb{R}^1$, and $w$ is the model parameter, our goal is to estimate a nontrivial gradient $\nabla_X f(X;w) \in \mathbb{R}^d$ with $\nabla_i f(X;w) \geq \theta$ or $\nabla_i f(X;w) = 0$, for all $i \in \{1, 2, \cdots, d\}$, and $\theta > 0$ is a threshold value for nontrivial gradient. This means we can ignore the gradients of dimensions whose gradients are smaller than $\theta$. First, we randomly sample $k$ dimensions in $X$, and get a set of dimensions $P = \{P_1, P_2, \cdots, P_k\}$, and use FD to estimate the gradients for dimensions in $P$. Then we sample a set of dimensions $P' = \{j \in P; \nabla_j f(X;w) \geq \theta\}$, and we use FD to estimate the gradients of the neighbors (a set $P''$) of dimensions in $P'$, if these gradients haven't been estimated. Then again we select dimensions with gradients no less than $\theta$ from $P''$ and find their neighbors to estimate gradients. We repeat this process for $n$ iterations. By exploring the sparsity of gradients, we can adaptively sample dimensions to estimate gradients, and can significantly reduce the amount of queries. We give an algorithm description of this attack obs-sfd-bb (which works in the same scenario as obs-fd-bb) in Algorithm 1 in the Appendix C. Here we provide an analysis of our SFD sampling algorithm and estimate the amount of non-trivial gradients that can be estimated using our method in Corollary 1. We will ignore the model parameter $w$ hereafter. The proof of the corollary is in Appendix C.1.

**Corollary 1.** *We make the following assumptions on $f$: 1) the gradient $\nabla f(x)$ satisfies: $|\nabla_i f(x) - \nabla_{i+1} f(x)| \leq \beta$ for all $i \in \{1, \cdots, d\}$ and for some $\beta > 0$; 2) suppose the current quantity to estimate is $\nabla_i f(x)$ and the next quantity to estimate is $\nabla_j f(x)$, where $|i - j| = 1$, we make the assumption on the gradient probability distribution among neighboring dimensions:*

$$P(|\nabla_j f(x)| \in [\theta, \beta + \theta]) = q, P(|\nabla_j f(x)| \in (\beta + \theta, \infty)) = 1 - q, \text{if } |\nabla_i f(x)| \in [\beta + \theta, \infty);$$
$$P(|\nabla_j f(x)| \in [0, \theta]) = q, P(|\nabla_j f(x)| \in [\theta, \beta + \theta]|) = 1 - q, \text{if } |\nabla_i f(x)| \in [\theta, \beta + \theta).$$

*Suppose within one iteration we estimate the gradient on dimension $j$: $\nabla_j f(x)$, denote $P(\theta)$ as the probability that $\nabla_j f(x) \geq \theta$ using our adaptive sampling method, and $P_{random}(\theta)$ as the same probability using random sampling, then we conclude: (1) $P(\theta) > 1 - q$; (2) if $1 - q > P_{random}(\theta)$, then $P(\theta) > P_{random}(\theta)$; (3) define $S_\theta = \sum_{i=1}^d 1(|\nabla_i f(x)| \geq \theta)$ and assume we estimated in total $S_\theta$ dimensions' gradients with perturbation strength $h$, then the truncation error of gradients estimation is upper bounded by the following inequality,*

$$\|\nabla \hat{f}(x) - \nabla f(x)\|_1 \leq S_\theta C h^2 + (d - S_\theta)\theta, \tag{1}$$

*where $C$ is some positive constant, and $\nabla \hat{f}(x)$ is the estimated gradient of $f$ with respect to $x$.*

### 4.1.3 ONLINE SEQUENTIAL ATTACKS

In a DRL setting, consecutive observations are not i.i.d.—instead, they are highly correlated, with each state depending on previous ones. It's then possible to perform an attack with less computation than perform the attack independently on each state. Considering real-world cases, for example, an autonomous robot would take a real-time video as input to help make decisions, an attacker is motivated to generate perturbations only based on previous states and apply it to future states, which we refer to as an *online sequential attack*. We hypothesize that a perturbation generated this way is effective on subsequent states.

**Universal attack based approach**. We propose online sequential attacks obs-seq-fgsm-wb, obs-seq-fd-bb, and obs-seq-sfd-bb that exploit this structure of the observations. obs-seq-fgsm-wb

works in standard white-box setting, where we know the architecture and parameters of the policy network; obs-seq-fd-bb and obs-seq-sfd-bb works in the same setting as obs-fd-bb. In these attacks, we first collect a number $k$ of observation frames and generate a single perturbation using the averaged gradient on these frames (or estimated gradients using FD or SFD, in the case of obs-seq-fd-bb and obs-seq-sfd-bb). Then, we apply that perturbation to all subsequent frames. In obs-seq-sfd-bb, we combine the universal attack approach and the adaptive sampling technique for finite difference estimates. We improve upon the above attack by finding the the set of frames that appear to be most important and using the gradients from those frames. With this, we hope to maintain attack effectiveness while reducing the number of queries needed. We propose to select a subset of frames within the first $k$ based on the variance of their $Q$ values. Then, in all subsequent frames, the attack applies a perturbation generated from the averaged gradient. We select an optimal set of important frames with high value variance to generate the perturbations. We give a proof in Corollary 2 below for why attacking these important frames is more effective, and we give the full proof in Appendix D.

**Corollary 2.** *Let the state and state-action value be $V(s)$ and $Q(s, a)$ respectively, and let the state with higher variance of Q value be state $s_{t_1}$ and the state with smaller variance of Q value be $s_{t_2}$. The variance is taken over different actions. Let the current policy be $\pi$. We have $\mathbb{E}_\pi \left[ \sum_{t=0}^T \gamma^t r_t | do(s_{t_1} = \hat{s}_{t_1}) \right] \leq \mathbb{E}_\pi \left[ \sum_{t=0}^T \gamma^t r_t | do(s_{t_2} = \hat{s}_{t_2}) \right]$, where $do(s_{t_1} = \hat{s}_{t_1})$ means the observation at time $t_1$ is changed from $s_{t_1}$ to $\hat{s}_{t_1}$.*

### 4.2 ATTACKS ON ACTION SELECTION

Our second category of attacks is to directly attack action output and minimize the expected return. We experiment with one attack in this category, under a white-box scenario, act-nn-wb. Here we train another policy network that takes in the state $s$ and outputs a perturbation on the $Q$ function: $Q'(s, a, w)$, the goal is also to minimize the expected return. For example, in DQN, the loss is chosen to be $L(w) = (Q(s, a) + Q'(s, a, w) - \tilde{r} - \gamma \max_{a'}(Q(s', a') + Q'(s', a', w)))^2$. For DDPG, the loss is chosen to be $L(w) = (Q(s, a = \mu(s)) + Q'(s, a = \mu(s), w) - \tilde{r} - \gamma(Q(s', a' = \mu(s')) + Q'(s', a' = \mu(s'), w)))^2$, where $\tilde{r} = -r$ is reward that captures the attacker's goal of minimizing the victim agent's expected return. This second approach to learn the attack $h$ is to treat the environment and the original policy $\pi$ together as a new environment, and view attacks as actions.

### 4.3 ATTACKS ON ENVIRONMENT DYNAMICS

In this third category, attacks perturb the environment transition model. In our case, we aim to achieve targeted attack, which means we want to change the dynamics such that the agent will fail in a specific way. Define the environment dynamics as $\mathcal{M}$, the agent's policy as $\pi$, the agent's state at step $t$ following the current policy under current dynamics as $s_t$, and define a mapping from $\pi, \mathcal{M}$ to $s_t$: $s_t \sim f(s_t | \pi, \mathcal{M}, s_0)$, which outputs the state at time step $t$: $s_t$ given initial state $s_0$, policy $\pi$, and environment dynamics $\mathcal{M}$. The task of attacking environment dynamics is to find another dynamics $\mathcal{M}'$ such that the agent will reach a target state $s'_t$ at step $t$: $\mathcal{M}' = \arg \min_\mathcal{M} \|s'_t - \mathbb{E}_{s_t \sim f(s_t | \pi, \mathcal{M}, s_0)}[s_t]\|$. **Random dynamics search**. A naive way to find the target dynamics, which we demonstrate in env-rand-bb, is to use random search. Specifically, we randomly propose a new dynamics and see whether, under this dynamics, the agent will reach $s'_t$. This method works in the setting where we don't need to have access to the policy network's architecture and parameters, but just need to query the network. **Adversarial dynamics search**. We design a more systematic algorithm based on RL to search for a dynamics to attack and call this method env-search-bb. At each time step, an attacker proposes a change to the current environment dynamics with some perturbation $\Delta \mathcal{M}$, where $\|\Delta \mathcal{M}/\mathcal{M}\|$ is bounded by some constant $\epsilon$, and we find the new state $s_{t, \mathcal{M}'}$ at time step $t$ following the current policy under dynamics $\mathcal{M}' = \mathcal{M} + \Delta \mathcal{M}$, then the attacker agent will get reward $\tilde{r} = 1/\|s_{t, \mathcal{M}'} - s'_t\|$. We demonstrate this in env-search-bb using DDPG (Lillicrap et al., 2016) to train the attacker. In order to show that this method works better than random search, we also compare with the random dynamics search method, and keep the bound of maximum perturbation $\|\Delta \mathcal{M}/\mathcal{M}\|$ the same. This attack works in the same setting as env-rand-bb.

## 5 EXPERIMENTS

We attack several agents trained for five different RL environments: Atari games Pong and Enduro (Bellemare et al., 2013), HalfCheetah and Hopper in MuJoCo (Todorov et al., 2012), and the driving simulation TORCS (Pan et al., 2017). We train DQN (Mnih et al., 2015) on Pong, Enduro and

TORCS, and we train DDPG (Lillicrap et al., 2016) on HalfCheetah and Hopper. The reward function for TORCS comes from Pan et al. (2017). The DQN network architecture comes from Mnih et al. (2015). The network for continuous control using DDPG comes from Dhariwal et al. (2017). For each game, we train the above agents with different random seeds and different architectures in order to evaluate different conditions in the transferability and imitation learning based black-box attack. Details of network structure and the performance for each game are included in Appendix A.

## 5.1 Experimental Design

We compare the agents' performance under all attacks with their performance under no attack, denoted as non-adv.

**Attacks on observation**. We test these attacks under $L_\infty$ perturbation bounds of $\epsilon = 0.005$ and $\epsilon = 0.01$ on the Atari games and MuJoCo simulations and $\epsilon = 0.05$ and $\epsilon = 0.1$ on TORCS [1]. First, we test the white-box attacks obs-fgsm-wb and obs-nn-wb on all five environments. Second, we test the attack obs-fgsm-bb under two different conditions: (1) In obs-fgsm-bb(1), the attacker uses the same network structure in the surrogate model as the victim policy and (2) In obs-fgsm-bb(2), the attacker uses a different network structure for the surrogate model. We test the attack obs-imi-bb on all five environments. Similar to the transferability attacks, we test this attack under same-architecture (obs-imi-bb(1) and different-architecture (obs-imi-bb(2)) conditions. We use FGSM to generate perturbations on the surrogate policy. We test obs-sfd-bb under different numbers of SFD iterations; we denote an attack that uses $i$ iterations as obs-s[$i$]fd-bb. The number of queries is significantly reduced in obs-sfd-bb than obs-fd-bb; we show the actual numbers of queries used in Table 4 in the Appendix. For the attack obs-seq-fgsm-wb, we test under the condition obs-seq[F$k$]-fgsm-wb (F for "first"), where we use *all* of the first $k$ frames to compute the gradient for generating a perturbation for the subsequent frames. For the attacks obs-seq-fd-bb and obs-seq-sfd-bb, we test under three conditions. (i) In obs-seq[F$k$]-fd-bb, we look at the first $k$ frames and use FD to estimate the gradient; (ii) In obs-seq[L$k$]-fd-bb and obs-seq[L$k$]-s[$i$]fd-bb (L for "largest"), we again look at the first $k$ frames, but we select only the $20\%$ of the frames that have the *largest Q* value variance to generate the universal perturbation; (iii) obs-seq[S$k$]-fd-bb (S for "smallest") is similar to the previous one, we select $20\%$ of the first $k$ frames that have the *smallest Q* value variance to generate the universal perturbation. We additionally test a random perturbation based online sequential attack obs-seq-rand-bb, where we take a sample from uniform random noise to generate a perturbation and apply on all frames. Although this attack does not consider the starting frames, we still test it under different conditions obs-seq[F$k$]-rand-bb, where we start adding the random perturbation after the $k$-th frame. This makes it consistent with the other online sequential attacks that apply their perturbation after the $k$th frame. **Attacks on action selection**. We test the action selection attack act-nn-wb on the Atari games, TORCS, and MuJoCo robotic control tasks. **Attacks environment dynamics**. We test the environment dynamics attacks env-rand-bb and env-search-bb on the MuJoCo environments and TORCS. In the tests on MuJoCo, we perturb the body mass and body inertia vector, which are in $\mathbb{R}^{32}$ and $\mathbb{R}^{20}$ in HalfCheetah and Hopper environments, respectively. In the tests on TORCS, we perturb the road friction coefficient and bump size, which is in $\mathbb{R}^{10}$.

## 5.2 Experimental Results

**Attacks on observation**. Figure 2a shows the results of the attacks on observations on TORCS, including all methods on attacking observations and the results of non-adv. We show the results on the Atari games and MuJoCo in Figure 6 and Figure 9 in the Appendix. On TORCS, our neural network based attack obs-nn-wb achieves better attack performance than the FGSM attack obs-fgsm-wb. Under a black-box setting, our proposed imitation learning based attacks obs-imi-bb(1), obs-imi-bb(2), and the FD based attack obs-fd-bb achieves better attack performance than the transferability based attacks obs-fgsm-bb(1) and obs-fgsm-bb(2).

Figures 2b and 2c compare the cumulative rewards among different black-box methods. These figures show that the policy is vulnerable to all of the black-box methods. Specifically, they show that obs-s[$i$]fd-bb can achieve similar performance to FD under each value of the perturbation bound $\epsilon$. In the Appendix, we provide the number of queries for using obs-sfd-bb and obs-fd-bb, and the results show that obs-sfd-bb uses significantly less queries (around 1000 to 6000) than obs-fd-bb (around 14,000) but achieves similar attack performance. The SFD method only samples part of the pixels to calculate gradient while the vanilla FD method requires gradient computation at all pixels.

---

[1] Values are in range [0,1]

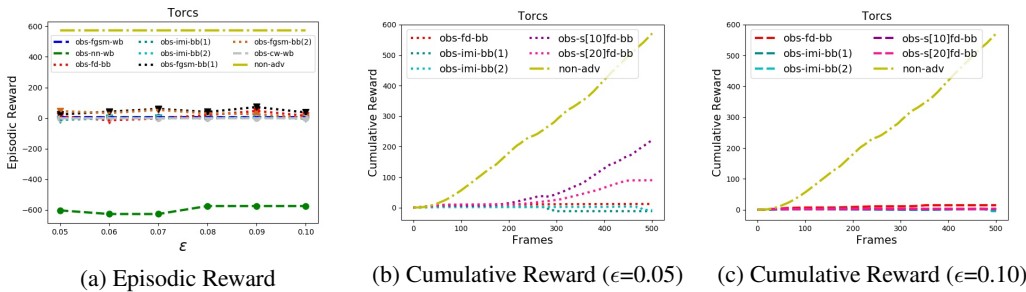

(a) Episodic Reward     (b) Cumulative Reward ($\epsilon$=0.05)     (c) Cumulative Reward ($\epsilon$=0.10)

Figure 2: Episodic reward under different attack methods and cumulative reward of different black-box attacks on TORCS.

Therefore, obs-sfd-bb is more efficient in terms of running time than obs-fd-bb, which indicates the effectiveness of our adaptive sampling algorithm in reducing gradient computation time and keeping the attack performance.

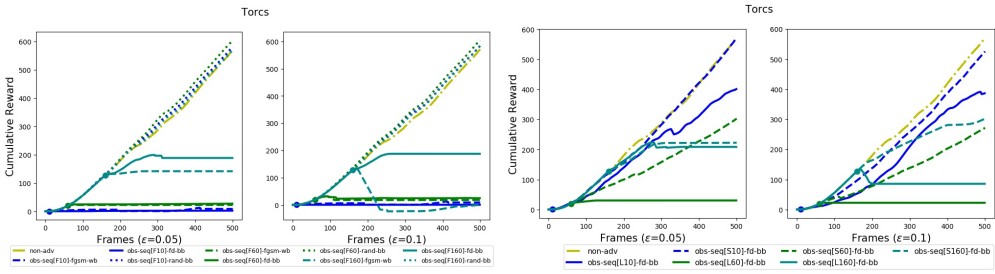

Figure 3: Performance of universal attack based approach considering all starting images ( (seq[F$k$]-, left two graphs) and subsets of frames with largest (seq[L$k$]-) and smallest (seq[S$k$]-) Q value variance (right two images). Results shown for TORCS, under two perturbation bounds $\epsilon$.

The results for comparing obs-seq[F$k$]-fgsm-wb, obs-seq[F$k$]-fd-bb, and obs-seq[F$k$]-rand-bb are shown in Figure 3 (left) for perturbation of different $L_\infty$ norm bound ($\epsilon = 0.05$ and $0.1$). The two figures show the cumulative reward for one episode when the states are under attack. Comparing the results, our proposed obs-seq[F$k$]-fd-bb achieves close attack performance compared with our obs-seq[F$k$]-fgsm-wb, and the baseline obs-seq[F$k$]-rand-bb is not effective. Figure 3 (right) shows that when we select a set of states with the largest Q value variance (obs-seq[L$k$]-fd-bb) to estimate the gradient, the attack is more effective than selecting states with the smallest Q value variance (obs-seq[S$k$]-fd-bb), which indicates that selecting frames with large Q value variance is more effective. We see that when $k$ is very small ($k = 10$), the estimated universal perturbation may be not accurate, and when $k = 60$, the attack performance is reasonably good.

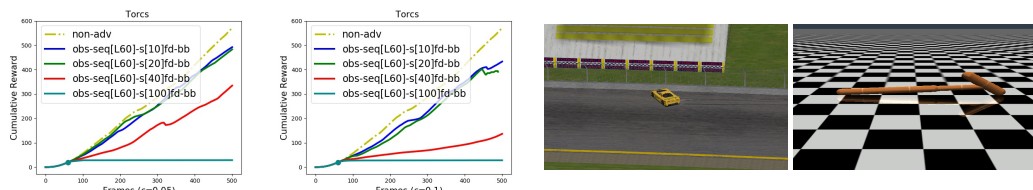

Figure 4: Performance of universal perturbation generated based on different numbers of query iterations with obs-seq-sfd-bb (left two graphs); final states of environment dynamics attack env-search-bb on TORCS and Hopper (right two images). TORCS image shows the top view of the driving environment when the car crashes.

In Figure 4 (left), we show the results of obs-seq[L$k$]-s[$i$]fd-bb by varying the number of iterations $i$, and select the 20% of frames with the largest Q value variance within the first $k$ frames to estimate the gradient using SFD. It is clear that with more iterations, we are able to get more accurate estimation of the gradients and thus achieve better attack performance, while the total number of queries is still

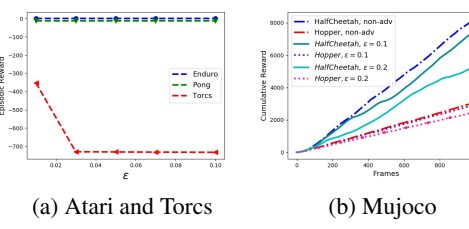

(a) Atari and Torcs  (b) Mujoco

Table 2: Environment Dynamics Attack Result.

| Environment | env-rand-bb | env-search-bb |
|---|---|---|
| HalfCheetah | 7.91 | 5.76 |
| Hopper | 1.89 | 0.0017 |
| TORCS | 25.02 | 22.75 |

Figure 5: Attack Action. Results of act-nn-wb on Atari games, TORCS, and MuJoCo tasks.

significantly reduced. We conclude in the Appendix that when $i = 100$, the number of queries for SFD is around 6k, which is significantly smaller than needed for FD, which takes 14k queries to estimate the gradient on an image of size $84 \times 84$ ($14112 = 84 \times 84 \times 2$). We put other results on attacks on observations in our Appendix part. The Enduro environment is also an autonomous driving environment that is simpler than the TORCS environment, and we observed consistent results in the two environments. Note that different thresholds $\epsilon$ are applied according to the complexity of the two environments.

**Attacks on action selection**. We present the results of our attacks on action selection in Figure 5. The results show that action space attack is also effective. With the larger perturbation bound, we achieve better attack performance.

**Attacks on environment dynamics**. In Table 2, we show our results for performing targeted adversarial environment dynamics attack. The results are the $L_2$ distance to the target state (the smaller the better). Our goal is to attack the environment dynamics so the victim agent will fail in a pre-specified way. For example, for a Hopper to turn over and for a self driving car to drive off road and hit obstacles. The results show that random search method performs worse than RL based search method in terms of reaching a specific state after certain steps. In Figure 4 (right), we show examples of the final state of our env-search-bb attack on TORCS and Hopper. In the final state of TORCS, the attacked car has hit the wall. The quality of the attack can be qualitatively evaluated by observing the sequence of states when the agent is being attacked and see whether the target state has been achieved. In Figures 11–13 in the Appendix, we show the sequences of states when the agents are under attack with the random search or reinforcement learning based search method.

## 6  DISCUSSION AND CONCLUSIONS

**Black-box and white-box**. As can be seen from our results, white-box methods' performance is better or on par with black-box methods. All white-box methods have almost similar performance and in some environments our proposed obs-nn-wb is better than obs-fgsm-wb. Within black-box methods, obs-fgsm-bb has slightly worse performance than obs-imi-bb. Finite difference methods' performance is worse than obs-imi-bb. Sampling based finite difference obs-sfd-bb is worse than obs-fd-bb and the performance improves when the number of query increases. In sequential attack based on optimal frame selection, selecting the frames with larger variance of Q-values to attack is better than selecting the frames with smaller variance of Q-values to attack, as can be seen from the results on TORCS.

**Connection with Robust RL**. There have been increasing interest in training RL algorithms that can be robust to perturbations in the environment, or even adversarial attacks in the environment. Previous methods that aim to improve the robustness of RL either try to apply some random perturbation to the observation or apply some gradient based noise to the observation to induce the agent to choose some sub-optimal actions. On the one hand, our finite difference and sampling based finite difference based method can provide faster attack than traditional FGSM based attack that requires back-propagation to calculate gradient, therefore can be incorporated into the training of RL policies to improve the robustness of RL policy. The environment dynamics attack can help to find the environment where the current agent is vulnerable. On the other hand, our methods provide tools to evaluate the vulnerability of the trained RL policy. Finally, we hope that our proposed taxonomy helps guide future research in making DRL systems robust, and we offer our experimental results as baselines for future robust RL techniques to compare against.

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

## A  EXPERIMENTAL SETUP

We trained DQN models on Pong, Enduro, and TORCS, and trained DDPG models on HalfCheetah and Hopper. The DQN model for training Pong and Enduro consists of 3 convolutional layers and 2 fully connected layers. The two network architectures differ in their number of filters. Specifically, the first network structure is $C(4, 32, 8, 4) - C(32, 64, 4, 2) - C(64, 64, 3, 1) - F(3136, 512) - F(512, na)$, where $C(c_1, c_2, k, s)$ denotes a convolutional layer of input channel number $c_1$, output channel number $c_2$, kernel size $k$, and stride $s$. $F(h_1, h_2)$ denotes a fully connected layer with input dimension $h_1$ and output dimension $h_2$, and $na$ is the number of actions in the environment. The DQN model for training TORCS consists of 3 convultional layers and 2 or 3 fully connected layers. The convultional layers' structure is $C(12, 32, 8, 4) - C(32, 64, 4, 2) - C(64, 64, 3, 1)$, and the fully connected layer structure is $F(3136, 512) - F(512, 9)$ for one model and $F(3136, 512) - F(512, 128) - F(128, 9)$ for the other model.

The DDPG model for training HalfCheetah and Hopper consists of several fully connected layers. We trained two different policy network structures on all MuJoCo environments. The first model's actor is a network of size $F(dim_{in}, 64) - F(64, 64) - F(64, na)$ and the critic is a network of size $F(dim_{in}, 64) - F(64, 64) - F(64, 1)$. The second model's actor is a network of size $F(dim_{in}, 64) - F(64, 64) - F(64, 64) - F(64, na)$, and the critic is a network of size $F(dim_{in}, 64) - F(64, 64) - F(64, 64) - F(64, 64) - F(64, 1)$. For both models, we added ReLU activation layers between these fully connected layers.

The TORCS autonomous driving environment is a discrete action space control environment with 9 actions, they are turn left, turn right, keep going, turn left and accelerate, turn right and accelerate, accelerate, turn left and decelerate, turn right and decelerate and decelerate. The other 4 games, Pong, Enduro, HalfCheetah, and Hopper are standard OpenAI gym environment.

The trained model's performance when tested without any attack is included in the following table 3.

Table 3: Model performance among different environments

|                 | Torcs  | Enduro | Pong | HalfCheetah | Hopper |
|-----------------|--------|--------|------|-------------|--------|
| Episodic reward | 1720.8 | 1308   | 21   | 8257        | 3061   |
| Episodic length | 1351   | 16634  | 1654 | 1000        | 1000   |

The DDPG neural network used for env-search-bb is the same as the first model (3-layer fully connected network) used for training the policy for HalfCheetah, except that the input dimension $dim_{in}$ is of the perturbation parameters' dimension, and output dimension is also of the perturbation parameters' dimension. For HalfCheetah, Hopper and TORCS, these input and output dimensions are 32, 20, and 10, respectively.

## B  DETAILS OF ADVERSARIAL ATTACKS ON DRL POLICIES

**Objective functions of the attack obs-nn-wb**. In obs-nn-wb, we train the parameters $w$ of the attacker network based on the given policy $\pi$ to minimize victim policy's expected return when the perturbations are applied: $w = \arg\max_w \mathbb{E}_{\pi(a|\tilde{s})}[\sum_t \gamma^t \tilde{r}_t] = \arg\max_w \mathbb{E}_{\pi(a|s+h(s,w))}[-\sum_t \gamma^t r_t]$. For example, in DQN, our goal is to perform gradient update on $w$ based on the following loss function: $L(w) = (Q(s + h(s, w), a) - (\tilde{r} + \gamma \max_{a'} Q(s' + h(s', w), a')))^2$, where $Q$ is the model under attack, $s'$ is the next state relative to current state $s$. In continuous control using DDPG, our goal is to perform gradient update on $w$ based on the following loss function $L(w) = (\tilde{r} + \gamma Q(s' + h(s', w), \mu(s' + h(s', w))) - Q(s + h(s, w), a))^2$, where $Q$ is the value function and $\mu$ is the actor function.

**Imitation learning based black-box attack details**. We provide the details of attack obs-imi-bb, which trains a surrogate policy to imitate the victim policy and apply the perturbation generated from the surrogate policy to the victim policy to perform the attack. Formally, in a Deep Q learning case, given a black-box policy $\pi^T$ with access to its policy outputs, we collect some dataset $\mathcal{D}^T = \{(s_i, \mathbf{q}_i)\}_{i=0}^N$, where each data sample consists of a short observation sequence $s_i$ and a vector $\mathbf{q}_i$ which is the unnormalized Q-values, and one value corresponds to one action. We will perform imitation learning to learn a new policy $\pi^S(\cdot|s, \theta)$ such that we minimize the following loss function by taking gradient update with respect to network parameters

$\theta$: $L(\mathcal{D}^T, \theta) = \sum_{i=1}^{|\mathcal{D}|} \text{softmax}(\frac{\mathbf{q}_i^T}{\tau}) \ln \frac{\text{softmax}(\frac{\mathbf{q}_i^T}{\tau})}{\text{softmax}(\mathbf{q}_i^S)}$, where $T$ corresponds to the victim policy, $S$ corresponds to our surrogate policy, and $\tau$ is a temperature factor. This attack works in the setting where we don't have access to the policy network's architecture or parameters, but can query the network.

**Finite difference based black-box attack details**. FD based attack on DRL uses FD to estimate gradient on the input observations, and then perform gradient descent to generate perturbations on the input observations. The key step in FD is to estimate the gradient. Denote the loss function as $L$ and state input as $\mathbf{s} \in \mathbb{R}^n$. Then the canonical basis vector $\mathbf{e}_i$ is defined as an $d$ dimension vector with 1 only in the $i$-th component and 0 otherwise. The finite difference method estimates gradients via the following equation

$$\mathbf{FD}(L(\mathbf{s}), \delta) = \left[ \frac{L(\mathbf{s} + \delta\mathbf{e}_1) - L(\mathbf{s} - \delta\mathbf{e}_1)}{2\delta}, \cdots, \frac{L(\mathbf{s} + \delta\mathbf{e}_d) - L(\mathbf{s} - \delta\mathbf{e}_d)}{2\delta} \right]^\mathsf{T}, \qquad (2)$$

where $\delta$ is a parameter to control estimation accuracy.

## C   FINITE DIFFERENCE GRADIENT ESTIMATION USING SFD

Algorithm 1 shows the detailed steps in our obs-sfd-bb attack. Table 4 shows the number of queries used in our experiments. We next provide a proof of proposed Corollary 1.

---

**Algorithm 1:** Adaptive sampling based finite difference (ASFD) algorithm

---

**input:**   $\mathbf{s} \in \mathbb{R}^d$: state vector;
        $f(w)$: loss function with parameter $w$;
        $k$: # of item to estimate gradient at each step ;
1         $n$: # of iteration;
        $\theta$: the gradient threshold used to filter out trivial gradients
        $\delta$: finite difference perturbation value ;
**output:** gradient $\nabla_{\mathbf{s}} f(\mathbf{s}; w)$;

2 **Initialization** :$\nabla_{\mathbf{s}} f(\mathbf{s}; w) \leftarrow \mathbf{0}$;
3 Randomly select $k$ indexes from $\{1, 2, \cdots, d\}$ to form an index set $P$ ;
4 **for** $t \leftarrow 0$ **to** $n$ **do**
5     **for** $j \in P$ **do**
6         **if** $\nabla_j f(\mathbf{s}; w)$ has not been estimated **then**
            /* Estimate gradient for position $j$         */;
7             Get $\mathbf{v} \in \mathbb{R}^d$ such that $\mathbf{v}_j = 1$ and $\mathbf{v}_i = 0, \forall i \neq j$;
8             Set $\nabla_j f(\mathbf{s}; w) = \frac{f(\mathbf{s}+\delta\mathbf{v};w) - f(\mathbf{s}-\delta\mathbf{v};w)}{2\delta}$ ;
9         **end**
10     **end**
11     $P' = \{j \in P; \nabla_j f(\mathbf{s}; w) \geq \theta\}$;
12     $P$ = indexes of neighbors of indexes in $P'$;
13 **end**
**Return:** $\nabla_{\mathbf{s}} f(\mathbf{s}; w)$

---

Table 4: Number of queries for SFD on each image among different settings. (The number of query for FD is 14112)

| bound | iteration | | | |
|---|---|---|---|---|
| | 10 | 20 | 40 | 100 |
| 0.05 | $1233 \pm 50$ | $2042 \pm 77$ | $3513 \pm 107$ | $5926 \pm 715$ |
| 0.10 | $1234 \pm 41$ | $2028 \pm 87$ | $3555 \pm 87$ | $6093 \pm 399$ |

### C.1   PROOF OF COROLLARY 1

*Proof.* Following the notation in Corollary 1, we use $P(\theta)$ to denote the probability that the to-be-estimated gradient magnitude of a dimension is no less than $\theta$ for some iteration. Initially, we sample

$k$ points uniformly. We have that out of the $k$ sampled points, there are $P(\beta + \theta) \cdot k$ points whose gradient magnitudes are no less than $\beta + \theta$. Similarly, we have $P(\theta) \cdot k$ points where the gradient magnitudes are no less than $\theta$. At the second iteration, we start from the $P(\theta) \cdot k$ points. According to the assumptions, we have $(1-q)P(\beta + \theta) \cdot k$ points whose gradient magnitudes are no less than $\beta + \theta$. Similarly, we have $q \cdot P(\beta + \theta) \cdot k + (1-q)[P(\theta) - P(\beta + \theta)]k$ points with gradient magnitudes in the range $[\theta, \beta + \theta]$. Now in $t^{th}$ iteration, we denote $a_t$ as the number of points in this iteration with gradient magnitudes no less than $\beta + \theta$ and $b_t$ as the number of points in this iteration with gradient magnitudes in range $[\theta, \beta + \theta]$. The recurrence relationship is as follows:

$$a_t = (1-q) \cdot a_{t-1}, \quad b_t = q \cdot a_{t-1} + (1-q) \cdot b_{t-1}, \tag{3}$$

where $a_1 = (1-q)k \cdot P(\beta + \theta)$ and $b_1 = (1-q)k \cdot P(\theta) + (2q-1)k \cdot P(\beta + \theta)$. We have:

$$\frac{b_t}{(1-q)^t} = \frac{q}{(1-q)^2} a_1 + \frac{b_{t-1}}{(1-q)^{t-1}},$$
$$\frac{b_t}{(1-q)^t} = \frac{q}{(1-q)^2} a_1 (t-1) + \frac{b_1}{(1-q)}. \tag{4}$$

Then we have:

$$a_t = (1-q)^{t-1} a_1, \quad b_t = (1-q)^{t-1} \cdot [(t-1)\frac{q}{1-q} \cdot a_1 + b_1] \tag{5}$$

Now we show that in each iteration, the probability that our algorithm samples dimensions with gradient magnitudes no less than $\theta$ is larger than that of random sampling ($P_{random}(\theta)$).

$$P(\theta) = \frac{a_t + b_t}{a_{t-1} + b_{t-1}} = (1-q) \cdot \frac{a_1 + (t-1) \cdot \frac{q}{1-q} \cdot a_1 + b_1}{a_1 + (t-2)\frac{q}{1-q} \cdot a_1 + b_1} > 1 - q. \tag{6}$$

Here from the previous iteration $t-1$ we obtained $a_{t-1} + b_{t-1}$ number of dimensions whose gradients have magnitude no less than $\theta$. From all of their neighbors, we obtained in probability $a_t + b_t$ number of dimensions with gradient magnitude no less than $\theta$. Therefore, $P(\theta) > 1 - q$. In other words, in each iteration, we have at least probability $1 - q$ to sample the points with gradient magnitudes no less than $\theta$. Now since $q$ measures the degree of steepness between two dimensions ( typically around $1/2$) and when the gradient distribution is sparse, which means a majority of pixels get very small gradients, which means $P_{random}(\theta)$ will be small, when we choose a relatively large $\theta$. Therefore as long as $1 - q > P_{random}(\theta)$, our sampling algorithm outperforms the random sample counterpart.

Now we prove the second part of our corollary: when $x \in \mathbb{R}^1$, assume function $f$ is $C^\infty$, by Taylor's series we have

$$f(x+h) = f(x) + f'(x)h + \frac{h^2}{2}f''(x) + \frac{h^3}{3!}f^{(3)}(x) + \cdots$$
$$f(x-h) = f(x) - f'(x)h + \frac{h^2}{2}f''(x) - \frac{h^3}{3!}f^{(3)}(x) + \cdots . \tag{7}$$

Combine the two equations we get

$$\frac{f(x+h) - f(x-h)}{2h} - f'(x) = \sum_{i=1}^{\infty} \frac{h^{2i}}{(2i+1)!}f^{(2i+1)}(x), \tag{8}$$

which means the truncation error is bounded by $O(h^2)$. Moreover, we have

$$\left| \frac{f(x+h) - f(x-h)}{2h} - f'(x) \right| \leq Ch^2, \tag{9}$$

where $C = \sup_{t \in [x-h_0, x+h_0]} \frac{f^{(3)}(t)}{6}$, and $0 < h \leq h_0$.

We can regard each dimension as a single variable function $f(x_i)$, then we have

$$\left| \frac{f(x_i+h) - f(x_i-h)}{2h} - f'(x_i) \right| \leq Ch^2. \tag{10}$$

For the dimensions where the gradient magnitudes are no less than $\theta$, from the above estimation, the total truncation error caused by these dimensions are no greater than $S_\theta Ch^2$. For those dimensions

where the gradient magnitudes are less than $\theta$, the error caused by not considering these dimensions are no greater than $(d - S_\theta)\theta$. All in all, the truncation error of gradients estimation is upper bounded by the following inequality.

$$\|\nabla \hat{f}(x) - \nabla f(x)\|_1 \leq S_\theta C h^2 + (d - S_\theta)\theta, \tag{11}$$

where $C$ is some positive constant, and $\nabla \hat{f}(x)$ is the estimated gradient of $f$ with respect to $x$.

$\square$

## D    PROOF OF COROLLARY 2

*Proof.*  Recall the definition of Q value is

$$Q(s_\tau, a_\tau) = \mathbb{E}_\pi[\sum_{t=\tau}^{H-1} \gamma^{t-\tau} r_t | s_\tau, a_\tau]. \tag{12}$$

The variance of Q value at a state $s$ is defined as

$$Var(Q(s)) = \frac{1}{|\mathcal{A}| - 1} \sum_{i=1}^{|\mathcal{A}|} \left(Q(s, a_i) - \frac{1}{|\mathcal{A}|} \sum_{j=1}^{|\mathcal{A}|} Q(s, a_j)\right)^2, \tag{13}$$

where $\mathcal{A}$ is the action space of the MDP, and $|\mathcal{A}|$ denotes the number of actions. Assume a fixed horizon of length $H$. Suppose we are to attack state $s_m$ and state $s_n$ where the Q value variance of this two states are $Var(Q(s_m))$ and $Var(Q(s_n))$, and assume $m < n$. Denote the state-action pair Q values after attack are $Q(s_m, \hat{a}_m)$ and $Q(s_n, \hat{a}_n)$, respectively. During the attack, state $s_m$ is modified to $\hat{s}_m$, and state $s_n$ is modified to $\hat{s}_n$, and their action's Q-value also change, so we use $\hat{a}_m$ and $\hat{a}_n$ to denote the actions after the attack. By using $s_m$ and $s_n$ instead of $\hat{s}_m$ and $\hat{s}_n$, we mean that though the observed states are modified by the attack algorithm, but the true states do not change. By using a different action notation, we mean that since the observed states have been modified, the optimal actions at the modified states can be different from the optimal actions at the original observed states. Then the total discounted expected return for the entire episode can be expressed as (assume all actions are optimal actions)

$$\begin{aligned} Q' &= Q(s_0, a_0) - \gamma^m Q(s_m, a_m) + \gamma^m Q(s_m, \hat{a}_m), \\ Q'' &= Q(s_0, a_0) - \gamma^n Q(s_n, a_n) + \gamma^n Q(s_n, \hat{a}_n). \end{aligned} \tag{14}$$

Since $m < n$, $Q''$ can also be expressed as

$$\begin{aligned} Q'' &= Q(s_0, a_0) - \gamma^m Q(s_m, a_m) + \gamma^m Q(s_m, a_m) \\ &\quad - \gamma^n Q(s_n, a_n) + \gamma^n Q(s_n, \hat{a}_n). \end{aligned} \tag{15}$$

Subtract $Q'$ by $Q''$ we get

$$\begin{aligned} Q' - Q'' &= \gamma^m(Q(s_m, \hat{a}_m) - Q(s_m, a_m)) + \gamma^n Q(s_n, a_n) - \gamma^n Q(s_n, \hat{a}_n) \\ &= -\gamma^m[Q(s_m, a_m) - Q(s_m, \hat{a}_m) - \gamma^{n-m}(Q(s_n, a_n) - Q(s_n, \hat{a}_n))]. \end{aligned} \tag{16}$$

According to our claim that states where the variance of Q value function is small will get better attack effect, suppose $Var(Q(s_m)) > Var(Q(s_n))$, and assume the range of Q value at step $m$ is larger than step $n$, then we have

$$\begin{aligned} Q(s_m, a_m) - Q(s_m, \hat{a}_m) &> Q(s_n, a_n) - Q(s_n, \hat{a}_n) \\ &> \gamma^{n-m}[Q(s_n, a_n) - Q(s_n, \hat{a}_n)]. \end{aligned} \tag{17}$$

Therefore $Q' - Q'' < 0$ which means attack state $m$ the agent will get less return in expectation. If $Var(Q(s_m)) < Var(Q(s_n))$, assume the range of Q value at step $m$ is smaller than step $n$, then we have

$$Q(s_m, a_m) - Q(s_m, \hat{a}_m) < Q(s_n, a_n) - Q(s_n, \hat{a}_n). \tag{18}$$

If $n - m$ is very small or $Q(s_n, a_n) - Q(s_n, \hat{a}_n)$ is large enough such that $Q(s_m, a_m) - Q(s_m, \hat{a}_m) < \gamma^{n-m}[Q(s_n, a_n) - Q(s_n, \hat{a}_n)]$, then we have $Q' - Q'' > 0$ which means attacking state $m$ the agent will get more reward in expectation than attacking state $n$. $\square$

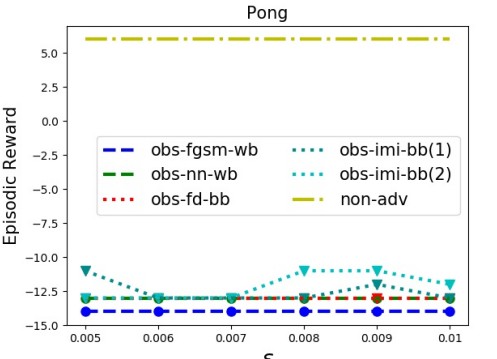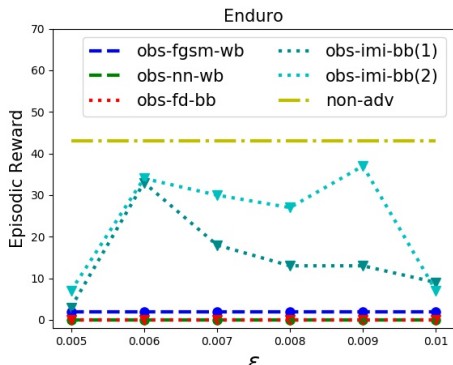

Figure 6: Episodic rewards among different attack methods on Atari games. Dotted lines are black-box attack while dash lines are white-box attack.

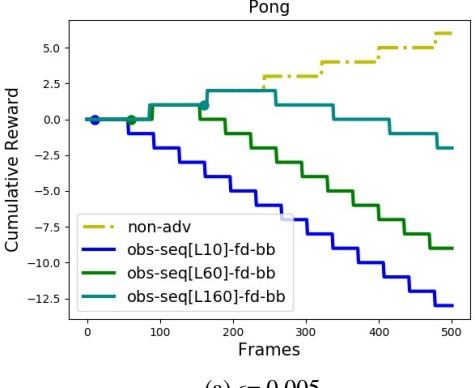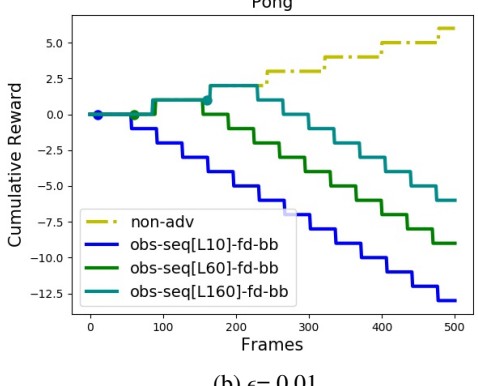

(a) $\epsilon = 0.005$                    (b) $\epsilon = 0.01$

Figure 7: Cumulative reward after adding optimal state based universal perturbation on Pong game

## E    RESULTS FOR ATTACK OBSERVATIONS IN OTHER ENVIRONMENTS

We provide the results of attack applied on observation space in Figure 6, Figure 7, Figure 8, Figure 9, and Figure 10. It can be observed from these results that, for obs-seq[L$k$]-fd-bb, there exists at least one $k > 0$ such that when we estimate a universal perturbation from the top 20% frames of the first $k$ frames and apply the perturbation on all subsequent frames starting from the $k$-th frame, we are able to achieve reasonably good attack performance. In some environments, such as in Pong, $k = 10$ is already enough to induce strong attack; while in Enduro, $k = 160$ achieves better performance than $k = 10$ or $k = 60$.

## F    RESULTS FOR DYNAMICS ATTACK

We include here the environment rollout sequence for dynamics attack experiment in Figure 11, Figure 12 and Figure 13. The last image in each sequence denotes the state at same step $t$. The last image in each abnormal dynamics rollout sequence corresponds to the target state, the last image in the attacked dynamics using RL search denotes the attacked results using env-search-bb, and the last image in the attacked dynamics using random search denotes the attacked results using env-rand-bb. It can be seen from these figures that env-search-bb method is very effective at achieving targeted attack while using random search, it is relatively hard to achieve this.

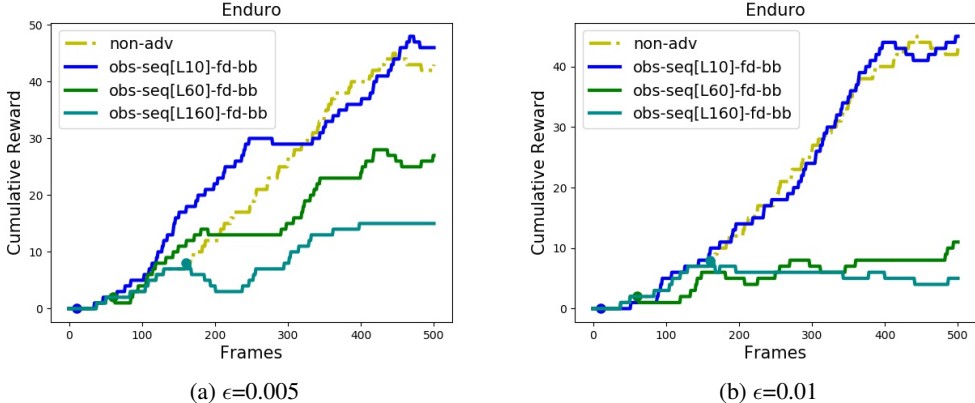

(a) $\epsilon$=0.005                    (b) $\epsilon$=0.01

Figure 8: Cumulative reward after adding optimal state based universal perturbation on Enduro game. The results are different from the TORCS results since the threshold $\epsilon$ is different from the TORCS case.

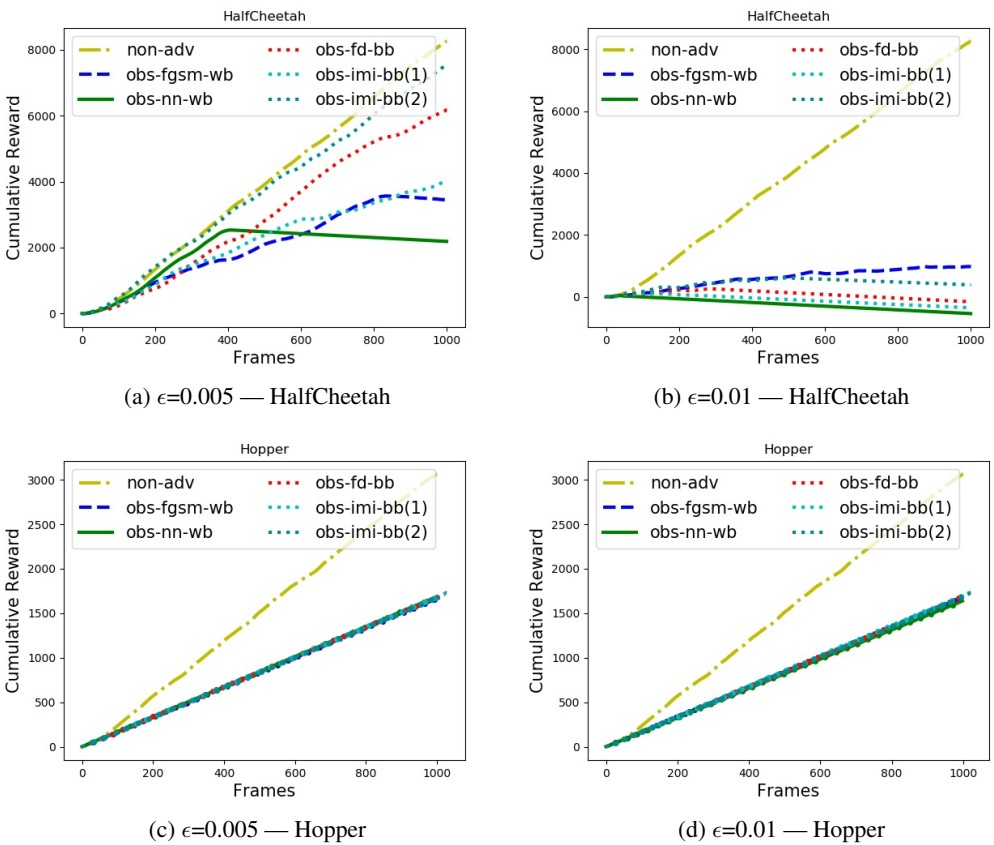

(a) $\epsilon$=0.005 — HalfCheetah            (b) $\epsilon$=0.01 — HalfCheetah

(c) $\epsilon$=0.005 — Hopper                 (d) $\epsilon$=0.01 — Hopper

Figure 9: Performance among different attack methods on MuJoCo. We use the format "$L_\infty$ bound—Environment" to label the settings of each image.

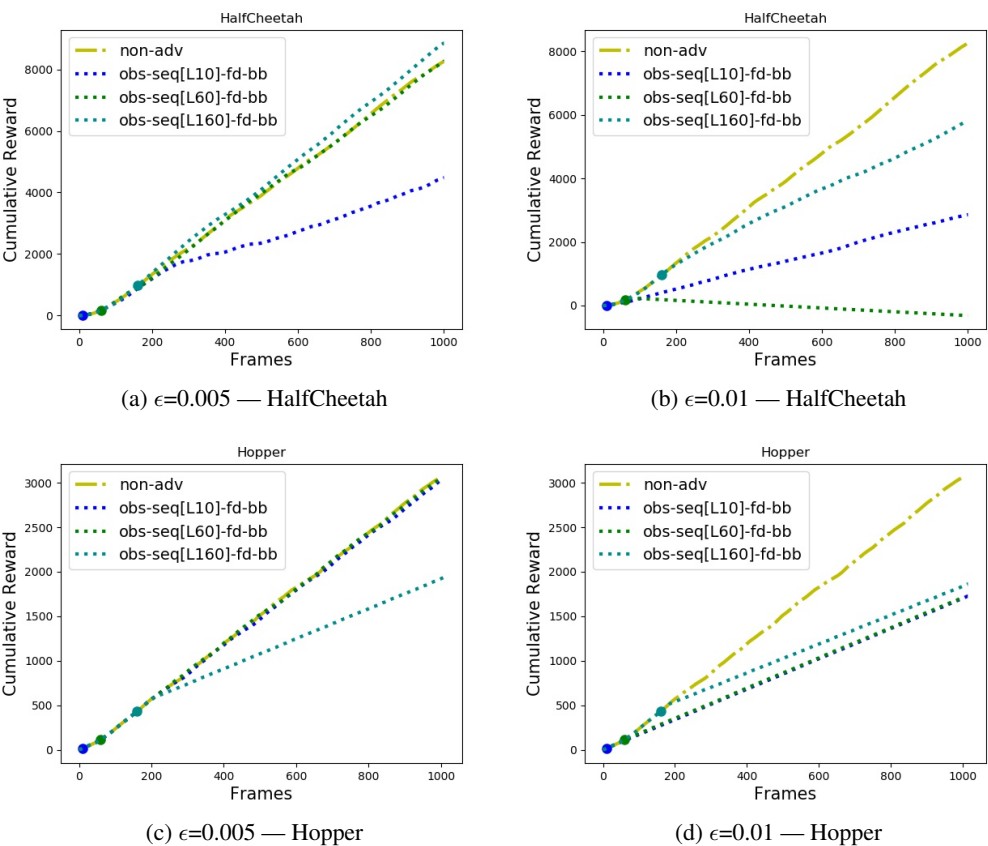

Figure 10: Cumulative reward after adding optimal state based universal perturbation on Mujoco. We use the format "$L_\infty$ bound— Environment" to label the settings of each image.

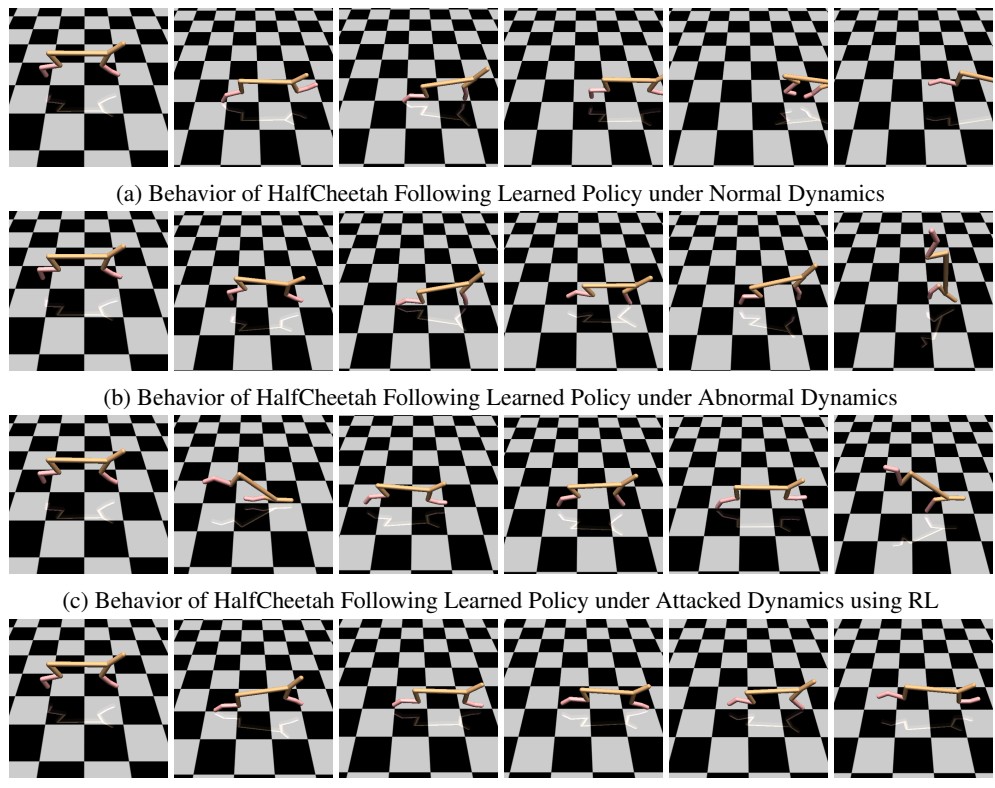

(a) Behavior of HalfCheetah Following Learned Policy under Normal Dynamics

(b) Behavior of HalfCheetah Following Learned Policy under Abnormal Dynamics

(c) Behavior of HalfCheetah Following Learned Policy under Attacked Dynamics using RL

(d) Behavior of HalfCheetah Following Learned Policy under Attacked Dynamics using Random Search

Figure 11: Results for Dynamics Attack on HalfCheetah

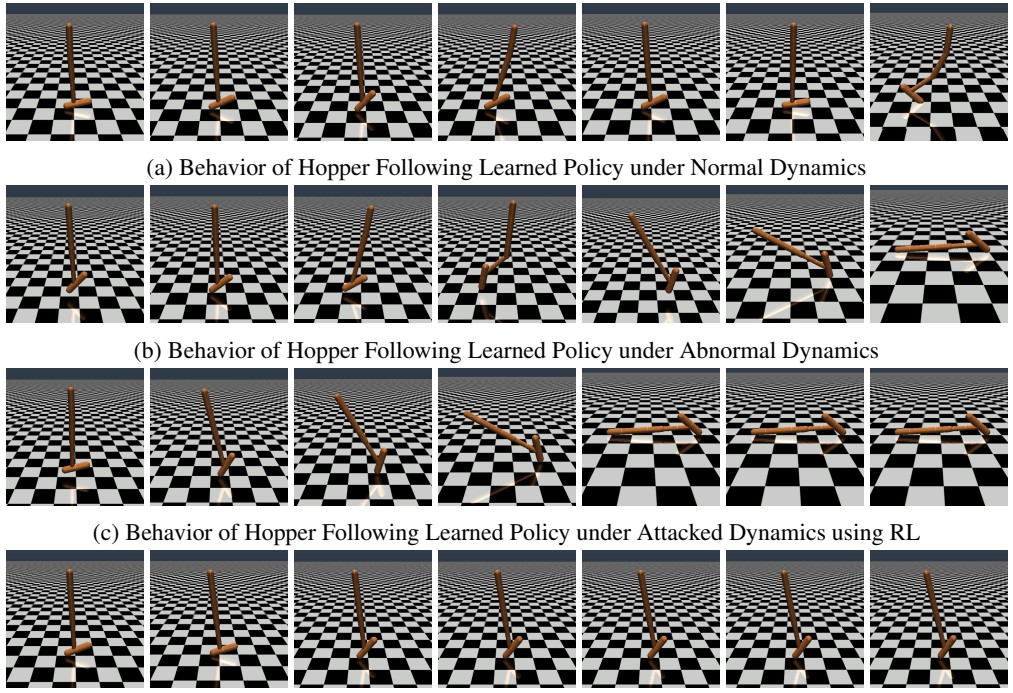

(a) Behavior of Hopper Following Learned Policy under Normal Dynamics

(b) Behavior of Hopper Following Learned Policy under Abnormal Dynamics

(c) Behavior of Hopper Following Learned Policy under Attacked Dynamics using RL

(d) Behavior of Hopper Following Learned Policy under Attacked Dynamics using Random Search

Figure 12: Results for Dynamics Attack on Hopper

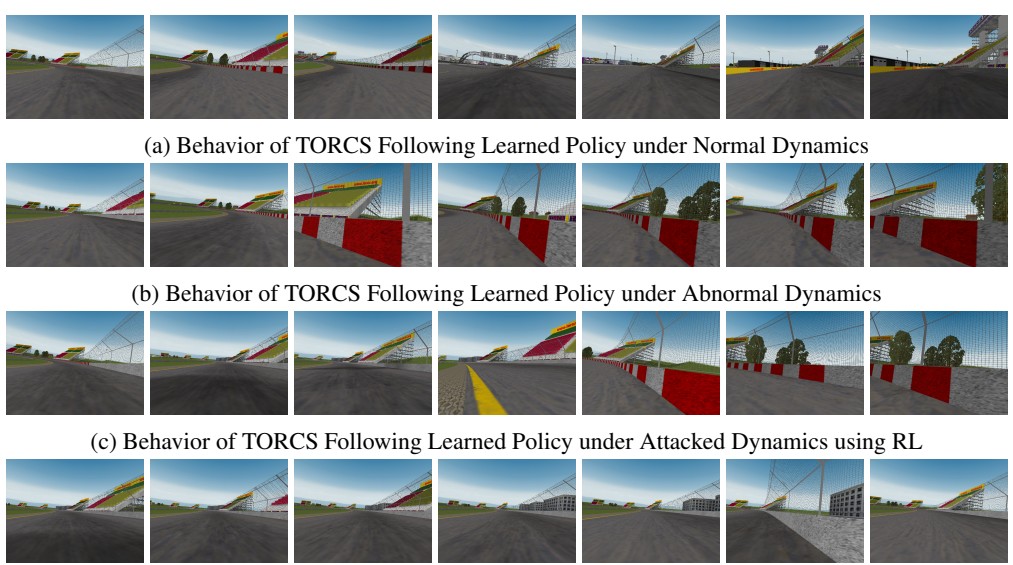

(a) Behavior of TORCS Following Learned Policy under Normal Dynamics

(b) Behavior of TORCS Following Learned Policy under Abnormal Dynamics

(c) Behavior of TORCS Following Learned Policy under Attacked Dynamics using RL

(d) Behavior of TORCS Following Learned Policy under Attacked Dynamics using Random Search

Figure 13: Results for Dynamics Attack on TORCS

