# OpenReview forum: "Characterizing Attacks on Deep Reinforcement Learning"
_ICLR.cc/2019/Conference_

### Official Review · AnonReviewer3 · 2018-11-02

**Rating:** 6
**Confidence:** 3

**Review:**

This submission sets out to taxonomize evasion-time attacks against deep RL and introduce several new attacks, including two heuristics for efficient evasion-time attacks and attacks that target the environment dynamics and RL system’s actions. The main limitation of this paper is probably its broad scope, which unfortunately prevents it in its current form from addressing each of the goals stated in the introduction systematically to draw conclusive takeaways.

Taxonomizing the space of adversaries targeting deep RL at test time is a valuable contribution. While the existing taxonomy is a good start, it would be useful if you can clarify the following points in your rebuttal. Why were the “further categorization” items separated from adversarial capabilities? Being constrained to real-time or physical perturbations appears to be another way to describe the adversary’s capabilities. In addition, is there a finer-grained way to characterize the adversary’s knowledge beyond white-box vs. black-box? This binary perspective is common but not very informative. One way to move forward would be for instance to think about the different components of a RL system, and identify those that are relevant to have knowledge of when adversaries are mounting attacks. It would also be helpful to position prior work in the taxonomy. Finally, the taxonomy currently stated in the submissions is more a taxonomy of attacks (or adversaries) than a taxonomy of vulnerabilities, so the title of Section 3 could perhaps be updated accordingly.

Section 4.1 gives a good overview of different attack strategies against RL based on modifying the observations analyzed by the agent. Many of these attacks are applications of known attack strategies and will be familiar to readers with adversarial ML background (albeit some of these strategies were previously introduced and evaluated against “supervised” classifiers only). One point was unclear however: why is the imitation learning based black-box attack not a transferability-based attack? As far as I could understand, the strategy described corresponds exactly to the commonly adopted strategy of transferring adversarial examples found on a substitute model (see for instance “Intriguing properties of neural networks” by Szegedy et al. and “Practical Black-Box Attacks against Machine Learning” by Papernot et al.). In other words, Section 4.1 could be rescoped to put emphasis on the attack strategies that have not been explored previously in the context of reinforcement learning: e.g., the finite difference approach with adaptive sampling or the universal attack with optimal selection of initial frames. It is unfortunate that the treatment of these two attacks is currently deferred to the appendix as they make the paper more informative. Similarly, Sections 4.2 and 4.3 would benefit from being extended to put forward the new attack threat model considered in these two sections.

While the introduction claimed to make a systematic evaluation of attacks against RL, the presentation of the experimental section can be improved to ensure the analysis points out the relevant takeaways. For instance, it is unclear what the differences are between results on TORCS and other tasks included in the Appendix. Specifically, results on Enduro do not seem as conclusive as those presented on TORCS. Do you have some intuition as to why that is the case? In Figure 7, it appears that a large number of frames need to be manipulated before a drop on cumulative reward is noticeable. Previous efforts manipulated single frames only, could you stress why the setting is different here? Throughout the section, many Figures are small and it is difficult to infer whether the difference between the white-box and black-box variants of an attack is significant or not. Could you analyze this in more details in the text? In Table 2, how should the L2 distance be interpreted? In other words, when is the adversary successful?

If you can clarify any of the points made above in your rebuttal, I am of course open to revise my review.

Editorial details:
Figures are not readable when printed.
Figure 5 is improperly referenced in the main body of the paper.
Figure 7: label is incorrect for Torcs and Hopper (top of figure)

---

> ### Author Response · Authors · 2018-11-26
> **To reviewer 3**
>
> Thanks for the useful feedback for our paper!
> Q: the scope is too broad which prevents it in its current form from addressing each of the goals stated in the introduction systematically to draw conclusive takeaways
> A: We’ve reorganized the paper and we try to focus on our proposed methods, and just mention the previously developed methods. For each of our proposed methods, we now discuss the realistic settings where these methods can be applied in section 3.
>
> Q: Why were the “further categorization” items separated from adversarial capabilities? And provide finer-grained way to characterize the adversary’s knowledge beyond white-box v.s. Black-box? Also talk about prior work in taxonomy.
> A: Sorry for the confusion of further categorization. We realize that adversarial capability is not an appropriate phrase to describe the first layer of categorization so we use the component of the MDP to be attacked. Then we provide further categorization on black-box and white-box method. Specifically, we classify black-box and white-box methods based on the detailed knowledge of the attacker has on the victim policy: whether the attacker knows the policy network’s architecture, weight parameters, and whether the attacker can query the policy network. We also introduced prior work in generating adversarial attacks on RL.
>
> Q: The taxonomy currently stated in the submissions is more a taxonomy of attacks (or adversaries) than a taxonomy of vulnerabilities
> A: We changed section 3’s title to be “taxonomy of attacks on DRL”.
>
> Q: why is the imitation learning based black-box attack not a transferability-based attack?
> A: You are correct, that the imitation learning based attack is also a transferability attack. We continue to distinguish the case where the local model is distilled as opposed to trained separately, but we have removed the text that suggests that it is not a transferability attack.
> In addition, the transferability based attack requires access to the environment’s state space, action space, transition dynamics and reward function to train a new policy. However, our imitation learning based attack only requires access to the environment’s state, action space and transition dynamics. We do not learn the new policy using reinforcement learning so we do not need to know the reward function.
>
> Q: It is unfortunate that the treatment of the proposed new attacks is currently deferred to the appendix as they make the paper more informative
> A: We put more emphasis on our proposed attacks and put the mathematical analysis back in the paper.
>
> Q: Differences between results on different environments? Results on Enduro do not seem as conclusive as those presented on TORCS. Do you have some intuition as to why that is the case?
> A: We don’t observe much difference between the results on different environments per se. The reason we only put TORCS results in the paper is due to the limit of the number of allowed pages. The reason why the Enduro’s results seem to be different from TORCS results is that the attack perturbation threshold is larger on the TORCS environment, since TORCS is a more complex driving environment than Enduro, so in order to achieve reasonably good attack performance the perturbation strength will be larger. But this won’t affect our conclusion.
>
> Q: In Figure 7, it appears that a large number of frames need to be manipulated before a drop on cumulative reward is noticeable.
> A: The cumulative reward in figure 7 is aligned to the beginning of the episode, where no perturbation is applied. The teal lines, for example, represent experiments where the perturbation begins to be applied after frame 160. We mark where the perturbation begins to be applied with a dot, and in our experiments, the cumulative reward immediately diverges from the non-adversarial experiment. Previous experiments also didn’t only manipulate one image. Actually all images are perturbed after the first k frames, and the first k frames are used to estimate this universal perturbation.
>
> Q: Many figures are small, difficult to infer the difference between white-box and black-box variants is significant or not.
> A: We have added analysis in Section 6 about the difference between white-box and black-box results and what they indicate. We also tried to enlarge the figure size so that you can clearly see the difference. White-box methods have better or similar performance than black-box methods. However, white-box methods’ application scenario is very limited while black-box methods can be more powerful even we have limited knowledge about the victim policy.
>
> Q: In Table 2, how should the L2 distance be interpreted?
> A: In table 2, the L2 distance is a measure of the difference between the achieved state and the target state. We’ve clarified this in our revision. The measure is ad-hoc, but we also include a graphical example of an achieved state in figure 4. The adversary is successful if the target attack has qualitatively been achieved.

---

> > ### Comment · AnonReviewer3 · 2018-12-08
> > **re**
> >
> > Thank you for taking the time to write a response. I've increased my score by one to take into account changes described in the response. However, I would recommend making the taxonomy more specific to RL given that similar taxonomies were previously proposed in the context of classification. It would also help to have a better explanation for the difference between TORCS and Enduro: typically more complex problems have been found easier to attack (relatively---in terms of the perturbation magnitude), explaining why that is not the case here would be valuable.

---

> > > ### Author Response · Authors · 2018-12-17
> > > **Thanks for the further comments!**
> > >
> > > Thanks for the feedback! We have made the taxonomy to be specific to RL, namely, what kind of attacks may exist for specific components of MDP in RL environments. Traditionally, this attack only applies to neural network’s input observations. In the torcs environment, the road is relatively broader than the enduro environment, which makes it harder to attack since the vehicle would always have some space to allow some attacks and still remain stable. We also updated the paper here:
> > > https://drive.google.com/file/d/1pVSpI-q_vAOwaDS_Qz7enrPsVpdlohRp/view?usp=sharing

---

### Official Review · AnonReviewer2 · 2018-11-04
**interesting paper, unsure of experimental validation**

**Rating:** 5
**Confidence:** 4

**Review:**

The authors design a new taxonomy of attacks on deep RL agents - they developed three classes of attacks - attacks the modify the observation given to the agent, attacks that modify the action used by the agent and attacks that change the dynamics of the environment. In settings that have been studied previously, the authors show that they can find attacks more effectively than previous approaches can. They also study learning based and online attack generation approaches, that can be effectively used to quickly find adversarial attacks on the agent. The authors validate their approaches experimentally on Mujoco tasks and the TORCS driving simulator.

Quality: I found the paper's contributions difficult to understand - the significance of the three classes of attacks is not properly explained (in particular, I found the action perturbation to be difficult to justify in a real world setting). Further, the difficulty of generating attacks in each of these classes and the need for new algorithms is not explained properly. The need for effective ways to quickly generate adversarial attacks in RL is clear, but the authors' experiments don't seem to clarify that their proposed aproaches achieve this goal.

Clarity: The organization of section 4 makes the paper difficult to read - I would separate the taxonomy from the contribution of novel ways of generating adversarial attacks (the latter, imo, is the more significant contribution).

Originality: To the best of my knowledge, the authors propose novel kinds of attacks as well as novel attack algorithms on RL agent.

Significance: The problem considered is certainly significant. Despite the successes achieved by DeepRL, their robustness (in terms of distribution shifts, adversarial noise, model errors etc.) is of great importance when considering deploying these models. However, the presentation and experiments leave me unconvinced that the presented approaches are  a significant step ahead in attack generation (particularly in ways to generate attacks that can efficiently be incorporated into adversarial training of RL agents).

Cons
1. Unclear presentation of technical contributions, experimental results do not support the key contributions of faster attack generation
2. I am also unconvinced of the relevance of blackbox attack algorithms given the nascent stage of deepRL - since these agents are just being developed and their abilities need to improve significantly before they become deployable (and blackbox adversarial attacks are a real concern), I feel this work is premature and will need to be redone once more capable/robust agents can be trained for practical RL settings

###
In light of the revision, I have revised my score given the rewriting of section 3 that addresses the second con I raised above. However, due to the lack of clarity in presentation of the technical results in section 4 and the experiments in section 5, I feel that the paper still require improvement before it can be accepted.

---

> ### Author Response · Authors · 2018-11-26
> **To reviewer 2**
>
> We thank the reviewer for useful feedback.
> Q: the significance of the three classes of attacks is not properly explained, and the action perturbation is difficult to justify in a real world setting.
> A: the significance of the three classes of attacks is further discussed in section 3, where we now provide some realistic scenarios where each class of the proposed attacks can be applied. For attacks on observations or actions in a self driving car scenario, the attacker may induce the car to select some malicious actions to collide into obstacles such as pedestrians, which will cause severe traffic accidents. For attacks on environment dynamics, self driving car policy may fail if the dynamics are perturbed by changing some environment conditions, such as road friction coefficient. We now add discussions in section 3 about the action perturbation in real world setting. For example, when a robot is controlled remotely by a control center, the robot itself does not calculate the action locally, but the action signal is transmitted wirelessly from the control center via techniques like bluetooth. Therefore, interference on the bluetooth signal can be used to induce the robot to perform some malicious action. This is a real concern and we also add citations in the paper.
>
> Q: the difficulty of generating attacks in each of these classes and the need for new algorithms is not explained properly.
> A: generating white-box attacks has rigorous requirements for the victim policy: we have to know its architecture and parameters, which makes it difficult to apply attacks in real world settings, where we usually don’t have that access. Traditional black-box method such as transferability based attack does not always work pretty well and we have already shown this in the paper. The transferability based attack obs-fgsm-wb and imitation learning based attack obs-imi-bb all require access to the original training environment, which may not be practical in real world scenarios. So we need some new algorithms that can work in real world settings (not necessarily have access to the entire training environment, not necessarily have the original policy’s structure and parameters) and have reasonably good performance. Therefore, we propose black box attacks based on finite difference to estimate gradient, and several variants of FD based method that improves the efficiency of generating the attack so that the attack can be applied in real time.
>
> Q: clarify proposed attack achieves attacks faster than vanilla finite difference method?
> A: We now include this in our section 5.2 that sampling based finite difference method uses less number of query than vanilla finite difference method but achieves similar performance. Specifically, we provide the number of queries for the finite difference, sampling based finite difference method to show that our sampling based finite difference method achieves similar attack performance but uses significantly less number of queries. (around 50% deduction in the number of queries times).
>
> Q: Clarity of section 4: separate taxonomy from the contribution of novel ways of generating adversarial attacks
> A:  We now include the taxonomy in section 3, and in section 4, we mainly talk about our proposed attack methods
>
> Q: confused whether the proposed approaches are a significant step towards better attack generation, particularly how to make RL robust using our presented approaches
> A: We now add the discussion about how our proposed attacks can be useful for the adversarial training of RL agents in our section 6. For example, the black-box environment dynamics attack method can be incorporated into the training of RL agents to generate adversarial environment. Previous method such as Robust Adversarial Reinforcement Learning (RARL) (Pinto et al., ICML 2017) manually generates the attack while here we automatically generate environment dynamics attack using reinforcement learning.
>
> Q: The work is premature and will need to be redone once more robust agents are available in practical RL settings
> A: Recently, deep RL has been deployed in the real world, such as RL based computer games (AlphaGo), RL based robots, etc. Therefore, it’s very important to develop attack method to evaluate the robustness of the RL policies to ensure they can be safely deployed. However, current methods based on transferability still haves suboptimal performance and have strong assumptions about the knowledge of the victim policy, which may not be realistic in real world scenarios. Therefore, we proposed more novel and efficient black-box attack methods.

---

> > ### Comment · AnonReviewer2 · 2018-12-09
> > **thanks for the revision**
> >
> > I thank the authors for taking time to prepare a revision of the paper.
> >
> > Section 3 is indeed greatly improved and helps me better understand the significance of the various technical contributions presented.
> >
> > However, sections 4 and 5 are still difficult to parse and the significance of the technical and experimental results presented are still unclear. For example, in section 4, corollaries 1 and 2 are mathematical statements but are stated in informal language, making it difficult to understand the precise statements being made.
> >
> > In section 5, several attack algorithms are compared in terms of cumulative and epsiodic reward without explaining the significance of the reduction in reward in terms of the task being solved. For example, on Torcs, I would prefer to have seen a discussion of the frequency of crashes or other metrics relevant to the task at hand. Otherwise, the significance of the performance improvement in the attacks is difficult to evaluate.

---

> > > ### Author Response · Authors · 2018-12-17
> > > **Thanks for the further comments!**
> > >
> > > Thanks for the feedback! We tried to improve the corollary further using mathematical notation instead of informal prose. The improved version can be found in the updated paper here: https://drive.google.com/file/d/1pVSpI-q_vAOwaDS_Qz7enrPsVpdlohRp/view?usp=sharing
> > >
> > > Basically, the first corollary is to prove that our SFD method is more efficient in terms of estimating nontrivial gradients with absolute value no less than threshold theta, and it also comes up with a conclusion in terms of the truncation error upper bound our SFD can make using finite difference method and disregarding small gradients. Corollary 2 describes the conclusion that using our optimal frame based selection, we can achieve more efficient attack by attacking frames with larger variance of Q value than attacking frames with smaller variance of Q value.
> > >
> > > Discussion of the frequency of the crashes or other metrics relevant to the task: we tried to decompose the torcs reward into two parts, one is progress reward, namely, how many distance the vehicle traveled, the another one is the reward related with crash. We include the figure in the updated paper’s appendix. This figure shows the individual steps’ reward where the positive and negative rewards with absolute value between 0 and 2 are regular progress reward, and the -2.5 and -3.0 rewards correspond to catastrophe reward, we can see from this figure that our method is able to achieve significant catastrophe attack effects and the potential risk is severe.

---

### Official Review · AnonReviewer1 · 2018-11-05
**Connections of each attack setting to a specific threat scenario should be discussed.**

**Rating:** 5
**Confidence:** 4

**Review:**

The attack methods are clearly and extensively described and the paper is well organized overall. Some of the attacks are a straightforward variation of known attacks.  Strong original contributions are not found in this work while I do not think lack of original contributions is a minus for this type of paper. One concern is that the connections of each attack setting to a specific threat scenario in the real world are not discussed in this paper. The authors display 14 types of attacks under various settings. Which attack is likely to be performed by what kind of adversaries in what situation?  For this type of security research, contribution becomes weak without a connection to a threat in the real world. Suppose attack scenario A destroys a policy network more seriously than attack scenario B. Even in such a situation, a rescue for attack scenario A might not be needed if attack scenario A is not realistic at all. Even if connections to threats in the real world is not clear, it would be important for security analysis to learn about the worst case. Unfortunately, this work simply exhibits a catalogue of attacks against RL and does not give a deep insight into what we should do to make RL secure.

The summarization of the attack scenarios against RL is high quality and the results shown in this paper would be useful for many researchers. I expect authors to give more discussions on connections to the real world.

---

> ### Author Response · Authors · 2018-11-26
> **To reviewer 1**
>
> We thank the reviewer for useful feedback.
> Q: Lack of strong original contributions?
> A: We intend to contribute several original attacks that are applications of existing techniques. We have restructured our introduction to highlight these, from the revised list of contributions: (2) We propose two practical strategies, N-attack and online sequential attack, for performing real-time attacks on DRL systems; (3) We introduce the first attack that adversarially perturbs a DRL system’s environment dynamics, and the first attack that perturbs a DRL system’s actions; (5) We propose a method to select the optimal frames within consecutive frames to attack to achieve optimal attack effect and provide the mathematical analysis for that; and (6) We devise and evaluate 10 new attacks on several DRL environments. Note that only obs-fgsm-wb and obs-fgsm-bb are the existing attacks on DRL.
>
> Q: Connections of each attack setting to real world application scenario?
> A: We now add corresponding discussions in section 3 about how each category of the attacks within the taxonomy can be realistic in real world. In particular, we discuss the realistic scenarios of the attacks on observation, action space and attacks on environment dynamics. For attacks on observation space, the perturbations are added on observations such as images. When the agent learns the policy entirely in simulation, software virus can be installed into the rendering environments or the physical simulator such that the observed state can be modified. When the agent gets observations from real world, such as in autonomous model car, sometimes the observations are captured on the vehicle and then transmitted wirelessly to a remote computer to get the control signal. Some wireless communication techniques such as bluetooth are vulnerable to potential attacks. Therefore the pixel values can be modified by interfering the transmitted wireless signal. For attacks on action space, if the agent gets control signals wirelessly from some control center and does not compute the action locally, then the control signal can be modified by some attackers by interfering the transmitted wireless signal. Attacks on environmental dynamics can be achieved by changing the environment transition dynamics. For example, change the road surface friction coefficient will change the transition dynamics for autonomous driving car.
>
> Q: Which attack is likely to be performed by what kind of adversaries in what situation?
> A: When introducing each attack method, we now provide the settings these attacks are suitable for. For example, white-box attack obs-fgsm-wb requires full access to the policy network’s architecture and parameters and can query the network. We now also add these settings to table 1 under “attacker knowledge”, where we provide the information for each attack method whether it needs to have access to the policy network’s architecture, parameters or to query the network. In section 4, within the introduction for each attack, we also discuss these settings.
>
> Q: It’s important for security analysis to learn about the worst case. This work does not give a deep insight into what we should do to make RL secure.
> A: We agree that attack algorithm development should give insight about the worst case and insight about how to make RL secure. These proposed attack methods can be integrated into the training of RL policies to improve the policy’s robustness. For example, environment dynamics attack method can be used to perturb the training environment so as to find some challenging environments or find ways to cause catastrophe, and help to explore the worst case. These methods can also provide evaluation benchmarks to measure RL policies’ robustness once they are full trained. We now add a paragraph in section 6 discussing the connection of our work with robust RL training.

---

> > ### Comment · AnonReviewer1 · 2018-12-10
> > **Thanks for the revision.**
> >
> > After reading the revised manuscript, some of my concerns, especially in Q2, are resolved and the quality is improved. On the other hand, in this type of research, I think it is important to give ordering of priority for defense with respect to the risk caused by the attacks and likelihood of the attacks, which will give a strong message for the community of RL. The discussion for each attack is still made individually and need more discussions to learn what we should do to make RL secure. For this reason, I'd like to retain the evaluation score.

---

> > > ### Author Response · Authors · 2018-12-17
> > > **Thanks for the further comments!**
> > >
> > > Since this paper is more about attack instead of defense, due to the page limit, we only include a brief discussion about how to make rl more secure with our proposed attack methods. In terms of the ordering of priority for defense with respect to the risk caused by the attacks and the likelihood of the attacks, environment dynamics attack is easier to deploy than other attack methods, since it does not require us to make digital amendment of the observations or control signal given by the policy, it should be addressed first. Then the observation based attack and action based attack should be addressed, though deploying these attacks require some access to the policy networks’ software system. We added a new section in the paper section 6 about this. The new paper can be accessed here: https://drive.google.com/file/d/1pVSpI-q_vAOwaDS_Qz7enrPsVpdlohRp/view?usp=sharing

---

### Author Response · Authors · 2018-11-26
**Summary of Revision**

We thank the reviewers for their valuable comments and suggestions. Based on the reviews, we made the following update to our revision:
1. Since we analyzed several attacks (14) corresponding to different types based on our taxonomy and it is a bit hard to illustrate each of them, we reorganized most parts of our paper in the revision to make the taxonomy clearer. We also emphasized our main contributions in section 1, simplified the description of previous proposed attack and put more words on our contributions.
2. We made further categorization about white-box and black-box attack based on the detailed knowledge the attacker has about the victim policy, and updated table 1.
3. We added discussion about the connection between the proposed attacks against DRL and real-world scenarios in section 3, and we also added discussion and potential directions about how such vulnerability analysis on RL can help to build robust RL systems in section 6. We also discussed the settings in which each attack method is applicable when we introduce them.
4. We discussed in greater detail the proposed black-box attack algorithms’ properties and applicability of the proposed black-box attack to victim models of limited knowledge in section 4.1.2, and emphasize our contribution on the proposed method and made the corresponding analysis clearer in corollary 1 and corollary 2.
5. We added more results analysis, specifically, how different methods compare to each other in terms of their attack efficiency and performance and in terms of the difference of the knowledge required about the victim policy in section 5.2.

---

### Meta-Review · Area_Chair1 · 2018-12-15

**Confidence:** 5
**Recommendation:** Reject

**Metareview:**

The authors have delivered an extensive examination of deep RL attacks, placing them within a taxonomy, proposing new attacks, and giving empirical evidence to compare the effectiveness of the attacks. The reviewers and AC appreciate the broad effort, comprising 14 different attacks, and the well-written taxonomic discussion. However, the reviewers were concerned that the paper had significant problems with clarity of technical presentation and that the attacks were not well grounded in any sort of real world scenario. Although the authors addressed many concerns with their revision and rebuttal, the reviewers were not convinced. The AC believes that R1 ought to have increased their score given their comments and the resulting rebuttal, but the paper remains a borderline reject even with a corrected R1 score.